# Exploring variables associated with medication non-adherence in patients with type 2 diabetes mellitus

**Walid Al-Qerem**[1]*, **Anan S. Jarab**[2], **Mohammad Badinjki**[1], **Dana Hyassat**[3], **Raghda Qarqaz**[1]

**1** Department of pharmacy, Al-Zaytoonah University of Jordan, Amman, Jordan, **2** Faculty of Pharmacy, Department of Clinical Pharmacy, Jordan University of Science and Technology, Irbid, Jordan, **3** National Center for Diabetes, Endocrinology and Genetics, Amman, Jordan

* waleed.qirim@zuj.edu.jo

**Data Availability Statement:** https://doi.org/10.5281/zenodo.4461093.

## Abstract

### Objective

This study aims to assess medication adherence and explore its predictors in outpatients with type 2 diabetes.

### Method

This cross-sectional study collected socio-demographics, disease-related information, and different biomedical variables for type 2 diabetes patients attending a Jordanian Diabetes center. The four-item medication adherence scale (4-IMAS) and the beliefs about medications questionnaire (BMQ) which includes necessity and concerns were used. Stepwise backward quartile regression models were conducted to evaluate variables associated with the Necessity and Concerns scores. Stepwise ordinal regression was conducted to evaluate variables associated with adherence.

### Results

287 diabetic patients participated in the study. Almost half of the participants (46.5%) reported moderate adherence and 12.2% reported low adherence. Significant predictors of the adherence were necessity score (OR = 14.86, p <0.01), concern score (OR = 0.36, p <0.05), and frequency of medication administration (OR = 0.88, p- <0.01). Education was a significant predictor of Necessity and Concerns scores (β = 0.48, -0.2, respectively).

### Conclusion

Simplifying the medication regimen, emphasizing medication necessity and overcoming medication concerns should be targeted in future diabetes intervention programs to improve medication adherence and hence glycemic control among diabetic patients.

**Funding:** RQ received a fund from Al-Zaytoonah University of Jordan, grant number 22/23/2019-2020. https://www.zuj.edu.jo/ The funder had no role in study design, data collection and analysis, decision to publish, or preparation of the manuscript.

**Competing interests:** The authors have declared that no competing interests exist.

## Introduction

Diabetes mellitus is the most common endocrine chronic disease, affecting 5–10% of the general population [1]. It is predicted that the number of diabetic patients will reach 592 million by 2035 [2]. Because of the high morbidity and mortality rates from the disease, diabetes represents a significant economic burden in many countries [3,4]. Evidence-based studies indicate that glycemic control is one of the most important predictors of mortality and morbidity in patients with diabetes [5]. Epidemiological analysis of the United Kingdom Prospective Diabetes Study (UKPDS) revealed that a reduction in HbA1c% corresponds to a reduction in incidence of myocardial infarction, stroke, and microvascular complications [6]. However, despite the variety of available diabetes treatments, only 50% of diabetic patients can manage to achieve adequate glycemic control (HbA1c% <7%) [7]. In Jordan, only 34% of the diabetic patients who are visiting the National Center for Diabetes, Endocrinology and Genetics (NCDEG) had achieved the HbA1c% target [8].

One of the key contributors to glycemic control is medication adherence; which is defined by the World Health Organization as "the degree to which the person's behavior corresponds with the agreed recommendations from a health care provider" [9]. The causal relationship between glycemic control and medication adherence had been established by several studies [10–12]. Moreover, medication non-adherence raises the risk of diabetes complications which will also increase the burden of the disease and its management [10]. Several factors have been identified as associated with medication non-adherence in patients with different chronic diseases including type 2 diabetes. These factors include patients' sex, their beliefs about medications, disease duration, and education level [13,14]. Therefore, evaluating patients' level of adherence and exploring the factors associated with poor medication adherence should provide insight on implementing diabetes management intervention programs to improve patients' adherence and hence glycemic control among patients with type 2 diabetes. Hence, this study aims to evaluate medication adherence and its associated factors among outpatients with type 2 diabetes.

## Methods

### Study site and participants

A cross-sectional study was conducted. Participants were recruited from the National Center for Diabetes, Endocrinology and Genetics (NCDEG) outpatient clinic in Amman, Jordan. NCDEG is one of the largest diabetes clinics in Jordan providing services to more than 324,000 patients per year [15].the Services the clinic provides include diabetes education, dietary management, and diabetic foot care [15]. The enrolled participants were patients of 18 years of age or older who had been diagnosed with type 2 diabetes for one year or more, and were taking at least one anti-diabetic medication.

**Sample size calculation.** The equation used to compute the minimum required sample size to conduct ordinal regression is: required sample size = 50+8P, where p is the number of predictors. The study aimed to assess the association of different medication usage, age, gender education and income levels, necessity and concerns scores, and diabetes duration with adherence level. In total the study aimed to evaluate the association between 22 variables and adherence level therefore, the minimum required sample size is 226 subjects. However, the univariate analysis indicated that only three variables out of the evaluated 22 were significantly associated with adherence level and therefore, only these three variables were included in the final ordinal regression model. .

**Study procedure.** Out of 360 patients who met the inclusion criteria and were approached and invited to participate by the physicians and the assigned researcher at the NCDEG diabetes clinics between January 2020 and December 2020, a total of 287 patients agreed to participate with a response rate of 79.7%. The participants signed an informed consent form that included all study details. The researcher explained the study objectives to the patients and assured them that participation was voluntary. No incentives were provided. Participants were interviewed at private rooms in NCDEG and each interview lasted for about 15 minutes. Study ethical approvals were obtained from Al-Zaytoonah ethical committee (ethical approval number 22/23/2019-2020) and NCDEG research ethics committee (ethical approval number 1/2020).

## Study instruments

**Socio-demographics.** The following socio-demographic data were collected: age, sex, household monthly average income, and education level. The average household monthly income in Jordan is 1000 JOD [16]; therefore, participants were categorized into low and high-income groups based on this figure. Participants were also categorized into a low educational group that includes those with diploma degree or less and high education group that includes those with bachelors or postgraduate degrees. Disease factors were reported such as duration of diabetes and HbA1c%, medication-related variables including types of medications, medications frequencies and cost, and the number prescribed medications were collected for each participant using patients' medical files in NCDEG.

**Beliefs about medicines questionnaire (BMQ)-specific.** The validated Arabic version of the BMQ-specific [17] (S3 and S4 Appendixes) was used to assess participants' positive beliefs represented as perception of medicines' necessity (Necessity statements) and participants' negative beliefs represented as concerns about the medicines (Concerns statements) [18]. Each part consisted of five statements of 5-point Likert scale. The responses for the statements were 1: "Strongly disagree", 2:"Disagree", 3:"Not certain", 4:"Agree" and 5:"Strongly disagree". Means for both parts were computed; these means were considered Necessity score and Concerns score.

**The 4-item medication adherence scale.** The validated Arabic version of the 4-IMAS [19,20] (S1 and S2 Appendixes) was used with "Yes" or "No" responses. The adherence score was the sum of the responses for the 4 items as "Yes" response was given 1 point and "No" was given zero points. The participants were divided into three groups: Low adherence for the participants with a score of three or more, moderate adherence group for those with one or two points, and high adherence group for those with a score of zero.

**Medication cost.** The total cost of diabetic medication was calculated based on the sum of the monthly price of each medication. The price list was obtained from the NCDEG.

**Pilot study.** Although the Arabic version of the questionnaires were previously validated. Nevertheless, the questionnaires (socio-demographic sheet, 4-IMAS, and BMQ-specific) were presented to 30 participants that met the inclusion criteria. The participants confirmed that all the questions were clear and comprehensible. The data of the subjects that participated in the pilot study were excluded from the final data.

**Statistical analysis.** SPSS Version 26 was used to analyze the data [21]. Categorical variables were presented as frequencies (%), while continuous variables were expresses as means (SD). Univariable analysis was conducted and variables with p-values less than <0.05 were included in the regression models. Internal consistency of 4-IMAS, Necessity and Concerns statements was evaluated using Cronbach's alpha. The normality of Concerns and Necessity scores were assessed, and normality assumption were not met, therefore stepwise quantile regression models were conducted to evaluate variables associated with the Necessity and

Concerns scores. A stepwise ordinal regression was conducted to evaluate variables associated with adherence level. The model included adherence level (low, moderate, or high adherence) as the dependent variable, while the independent variables include Necessity score, Concerns score, and medications frequency (variables that were significantly associated with adherence level in the Univariable analysis). Because of the high significant correlation between total medication cost, number of medication and medication frequency, only medication frequency was included in the model. Moreover, to maintain adequate cell count assumption; several medications with expected cell count ≤ 5 were removed. P-values of less than 0.05 were considered to be significant.

## Results

Two hundred and eighty-seven diabetic patients participated in the study with a response rate of 79.7%. As shown in Table 1, the mean age of the participants was 56(±14). More than half of the participants (54.7%) were females. The participants were divided into a low-income group (42.9%) and high-income group (57.1%). The majority of the participants (77.4%) had an education level of diploma or less.

Disease and medication characteristics are presented in Table 2. The duration of disease ranged between 1 year and 21 years with a mean of 3.04(±3.52). The mean of the number of medications that participants were on is 6.66(±3.36) and the highest number of the prescribed medications was twenty. Many of the participants were taking more than one type of antidiabetic medication and Metformin was the most taken medication (86.4%), followed by sulfonylurea (31.0%) and Dipeptidyl peptidase ((DDP-4) Inhibitors (28.9%). The most common prescribed non-antidiabetic medication was statin (39.4%).

The BMQ-specific statements' means are shown in Table 3. Internal consistencies for Necessity and Concerns statements were confirmed by Cronbach's alpha (0.93 and 0.81 respectively). The Quantile regression indicated that lower education level ($\beta = 0.2$, $p = 0.003$) and statin administration ($\beta = 0.2$, $p < 0.001$) were associated with increased concerns score, while increased education level ($\beta = 0.48$, $p = 0.001$) was associated with increased necessity score.

Participants' adherence levels and their responses to the 4-IMAS are shown in Table 4. The computed Cronbach's alpha indicated acceptable internal consistency (0.74). Most of the participants reported moderate adherence (46.7%), followed by high adherence (41.1%) and lastly low adherence (12.2%). The most common form of non-adherence was forgetfulness (57.1%) while stopping the medication when feeling worse was the least form of non-adherence (13.9%).

The mean cost of diabetic medications was 28.33 JOD (±29.13) and the maximum monthly cost was 160.64 JOD/month. SGLT-2 inhibitors drug class had the highest cost and Sulfonylurea drug class had the lowest.

**Table 1. Socio-demographics.**

|  |  | Frequency (%) or Mean (SD) |
|---|---|---|
| **Age** |  | 56(14) |
| **Gender** | **Female** | 157(54.7) |
|  | **Male** | 130(45.3) |
| **Household monthly average income** | **Low income** | 123(42.9) |
|  | **High income** | 164(57.1) |
| **Education level** | **Low education** | 222(77.4) |
|  | **High education** | 65(22.6) |

**Table 2. Disease and medications characteristics of the participants.**

|  |  | Frequency (%) or Mean (±SD) |
|---|---|---|
| **Diabetes duration** |  | 6.04(±3.52) |
| **Number of medications** |  | 6.66(±3.36) |
| **Diabetic of medications** | **Metformin** | 248(86.4) |
|  | **DDP4- Inhibitors** | 83(28.9) |
|  | **Sulfonylurea** | 89(31.0) |
|  | **SGLT2 Inhibitors** | 13(4.5) |
|  | **Insulin** | 105(36.6) |
| **Other Medications** | **Statins** | 113(39.4) |
|  | **Beta Blockers** | 47(16.4) |
|  | **Nitrates** | 8(2.8) |
|  | **Gabapentin** | 11(3.9) |
|  | **Diuretics** | 27(9.5) |
|  | **ARBs** | 55(19.3) |
|  | **CCB** | 26(9.1) |
|  | **ACEI** | 14(4.9) |
|  | **PPI** | 57(19.9) |

DDP4: Dipeptidyl peptidase-4, SGLT2: Sodium-Glucose Cotransporter-2, ARBs: Angiotensin Receptor Blocker, CCB: Calcium Channel Blocker, ACEI: Angiotensin Convertor Enzyme Inhibitor, PPI: Proton Pump Inhibitor.

As shown in Table 5, results of ordinal regression revealed that increasing medications frequency (OR = 0.88, p-value<0.05) and increased medication concerns score (OR = 0.36, p-value <0.01) were associated with decreased medication adherence, while increased necessity score was significantly associated with increased medication adherence (OR = 14.86, p-value <0.01).

## Discussion

Diabetes medications including oral and injectable hypoglycemic agents are highly effective in disease management if taken properly [22]. Studies have shown that diabetic patients' adherence to oral hypoglycemic agents varies between 36% and 93% [23] and the overall adherence levels are greatly below average [24]. Among this study's participants, 12.2% were low

**Table 3. Beliefs about medicines questionnaire.**

|  | Mean(±SD) |
|---|---|
| **Specific Necessity** | **3.55(±0.79)** |
| My health, at present, depends on my medicines | 3.52(±0.91) |
| My life would be impossible without my medicines | 3.46(±0.92) |
| Without my medicines I would become very ill | 3.61(±0.85) |
| My health in the future will depend on my medicines | 3.56(±0.90) |
| My medicines protect me from becoming worse | 3.60(±0.87) |
| **Specific Concerns** | **3.08(±0.49)** |
| Having to take medicines worries me | 3.16(±0.61) |
| I sometimes worry about the long-term effects of my medicines | 3.08(±0.75) |
| My medicines are a mystery to me | 3.02(±0.64) |
| My medicines disrupt my life | 3.00(±0.49) |
| I sometimes worry about becoming too dependent on my medicines | 3.13(±0.72) |

**Table 4. The 4-item medication adherence scale.**

| Adherence statements | | Mean (±SD) or frequency (%) |
|---|---|---|
| Ever forget to take medicines (Yes) | | 164(57.1) |
| Ever careless about taking medicines (Yes) | | 56(19.5) |
| Stop taking medicines when feeling better (Yes) | | 44(15.3) |
| Stop taking medicines if you feel worse (Yes) | | 40(13.9) |
| Total score | | 1.06(±1.20) |
| Adherence level | | Frequency (%) |
| Low | | 35(12.2) |
| Moderate | | 134(46.7) |
| High | | 118(41.1) |

adherent, this percentage is lower than the ones reported among the diabetic patients' in studies conducted in Nigeria [25], USA [26], Palestine [27], and Saudi Arabia [28]. However, there is still a room for improvement as for instance, a study conducted in Ghana reported a percentage of 8% of low adherence level in participants [29]. These variations in reported adherence levels between different studies may be attributed to differences in methodologies (i.e., questionnaires used) and sample characteristics. The threshold for acceptable medication non-adherence among diabetic patients is yet to determine, but a French crowdsourcing study reported that according to medical doctors' opinions, one daily missed dose per month of insulin therapy is considered unacceptable [30]. Therefore, in addition to poor adherence, moderate adherence among diabetic patients also should be improved. A study that evaluated medication adherence in Jordanian patients with different chronic diseases who were taking five medications or more found that 46.1.% of the participants reported low adherence [31]. Higher rate of non—adherence (72.5%) was reported among outpatients with type 2 diabetes in Jordan [32]. The current study enrolled patients from a specialized diabetes center (NCDEG) and used additional instruments to determine the factors associated with adherence, including BMQ when compared with the earlier study by Jarab et al. [32].

Several modifiable and non-modifiable predictors of adherence level were investigated in the literature. A number of non-modifiable predictors had been found to be significantly but inconsistently associated with adherence level including sex and education level [26,33,34]. However, none of these variables was significantly associated with medication adherence in the present study. Nevertheless, education level was significantly associated with the participants' beliefs about medication which in turn was significantly associated with adherence level. Focus on modifiable predictors including factors related to behavioral aspects [35], costs [36], and therapy regimens complexity [37] has recently increased. Consistent with earlier research findings [25,27,38,39], patients' beliefs about diabetes medications was significantly associated with medication adherence in the present study. The same result was also observed in studies conducted on a sample of patients with different multiple chronic diseases [40], and

**Table 5. Ordinal regression of adherence level.**

| | B | Low adherence vs. High adherence | | | |
| | | P-value | Odds Ration | Confidence Interval of 95% | |
| | | | | Lower | Upper |
|---|---|---|---|---|---|
| Medications frequency | -0.13 | 0.032 | 0.88 | 0.79 | 0.99 |
| Necessity score | 2.70 | <0.01 | 14.86 | 0.20 | 0.67 |
| Concerns score | -1.02 | <0.01 | 0.36 | 8.69 | 25.42 |

on those with specific diseases such as cardiovascular diseases [41] dyslipidemia [17], and asthma [42]. In the current study, better patients' medication beliefs, represented as increased perception of the necessity of their medication, was associated with improved medication adherence. The role of healthcare providers in this aspect can be substantial where they can emphasize the important role of medications in improving disease management and health outcomes, which in turn motivates the patients to take the medications as recommended.

Increased patients' concerns about their medications was found to negatively impact medication adherence in the present study, as well as an unwillingness to initiate new therapies [43,44]. Therefore, exploring patient's concerns about their medications and resolving any potential barriers to take the medication as prescribed should be considered in future patient-centered interventions aimed at improving medication adherence and hence glycemic control and health outcomes among patients with type 2 diabetes. Moreover, team-based care that includes pharmacists had been found as an effective strategy to give patients opportunities to raise their concerns, which in turn, improves their adherence [45,46].

Due to the strong association between medication adherence and patients' beliefs about medications observed in the current study, it is necessary to investigate the variables which are associated with medication beliefs. Similar to a previous study [47], lower education level was associated with lower perception of medication necessity and increased concerns about the medications. Therefore, exploring the benefits of diabetes medications in controlling blood glucose and the serious complications which could be developed due to medication non-adherence should be explored for patients with a low level of education, particularly in during the delivery of disease management intervention programs. Similar to the current study findings, previous studies found that the administration of statins, particularly high-intensity statins, was associated with decreased medication adherence due to increased concerns related to potential adverse medication effects [48], this could lead to poor disease control and hence poor quality of life among statin recipients [49].

Consistent with the findings from a systemic review of 20 studies which all reported a negative impact of increased medication frequency on medication adherence [50], we also found that increased medication frequency was associated with decreased medication adherence in the present study. Simplifying dosage regimen by prescribing fixed-dose combinations [51] and medications with long half-lives should be considered in future diabetes management intervention programs.

## Limitations

Because some of the study results were based on self-reported data which were not independently confirmed, the results are subject to recall bias. Moreover, as the questionnaires were completed by the interviewer based on the participants' responses, the study results are subject to social desirability and interviewer bias, However, to reduce the effect of the interviewer bias each question was read exactly as it appears to the participants without paraphrasing or interpretation. Selection bias could be another limitation of the study as participants who were interested in the study objectives would be more encouraged to participate in the study. Moreover, the results of the current study are based on the data from one center only, however, the NCDEG is the only specialized center for diabetes in Jordan and receives patients from all over the kingdom.

## Conclusion

The current study findings show that there is area for medication adherence improvement among patients with type 2 diabetes. In addition to consider simple dosage regimen,

enhancing diabetes medication necessity and exploring medication concerns should be prioritized in future diabetes management intervention programs, particularly for patients with a lower level of educational.

## Supporting information

**S1 Appendix. Medication adherence questionnaire (Arabic).**
(DOCX)

**S2 Appendix. Medication adherence questionnaire (English).**
(DOCX)

**S3 Appendix. Beliefs about medications (BMQ)-general questionnaire (Arabic).**
(DOCX)

**S4 Appendix. Beliefs about medications (BMQ)-general questionnaire (English).**
(DOCX)

## Author Contributions

**Conceptualization:** Walid Al-Qerem, Mohammad Badinjki.

**Data curation:** Mohammad Badinjki, Raghda Qarqaz.

**Formal analysis:** Walid Al-Qerem, Raghda Qarqaz.

**Investigation:** Walid Al-Qerem, Anan S. Jarab, Dana Hyassat.

**Methodology:** Walid Al-Qerem, Anan S. Jarab.

**Project administration:** Walid Al-Qerem, Mohammad Badinjki, Dana Hyassat.

**Supervision:** Walid Al-Qerem.

**Writing – original draft:** Walid Al-Qerem, Mohammad Badinjki, Raghda Qarqaz.

**Writing – review & editing:** Walid Al-Qerem, Anan S. Jarab, Dana Hyassat, Raghda Qarqaz.

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
