## [Decision Letter · Decision Letter 0]

1 Jul 2021

PONE-D-21-17503

Exploring variables associated with medication non-adherence in patients with type 2 diabetes mellitus

PLOS ONE

Dear Dr. Al-Qerem,

Thank you for submitting your manuscript to PLOS ONE. After careful consideration, we feel that it has merit but does not fully meet PLOS ONE’s publication criteria as it currently stands. Therefore, we invite you to submit a revised version of the manuscript that addresses the points raised during the review process.

I have received the reports from our advisors on your manuscript which you submitted to PLOS ONE.

Based on the comments received, I feel that your manuscript could be reconsidered for publication should you be prepared to incorporate major revisions.

When preparing your revised manuscript, you are asked to carefully consider the reviewer comments below and submit a list of responses to the comments.

Editor Comments: The paper should be checked by a professional speaker of English before complete acceptance.

We look forward to receiving your revised manuscript.

Kind regards,

Muhammad Sajid Hamid Akash

Academic Editor

PLOS ONE

Journal Requirements:

3. Please include additional information regarding the survey or questionnaire used in the study and ensure that you have provided sufficient details that others could replicate the analyses. For instance, if you developed a questionnaire as part of this study and it is not under a copyright more restrictive than CC-BY, please include a copy, in both the original language and English, as Supporting Information. If the original language is written in non-Latin characters, for example Amharic, Chinese, or Korean, please use a file format that ensures these characters are visible.

4. Please state whether you validated the questionnaire prior to testing on study participants. Please provide details regarding the validation group within the methods section.

Reviewers' comments:

Reviewer's Responses to Questions

**Comments to the Author**

1. Is the manuscript technically sound, and do the data support the conclusions?

Reviewer #1: Yes

Reviewer #2: Yes

Reviewer #3: Partly

2. Has the statistical analysis been performed appropriately and rigorously? 

Reviewer #1: Yes

Reviewer #2: Yes

Reviewer #3: Yes

3. Have the authors made all data underlying the findings in their manuscript fully available?

Reviewer #1: Yes

Reviewer #2: Yes

Reviewer #3: Yes

4. Is the manuscript presented in an intelligible fashion and written in standard English?

Reviewer #1: Yes

Reviewer #2: Yes

Reviewer #3: Yes

5. Review Comments to the Author

Reviewer #1: Comments to editor and authors

Thank you for submitting this interesting manuscript that that evaluated the factors associated with non-adherence among diabetic Jordanians. The study is well written however, I have few comments that should be addressed.

-Major point:

Add sample size calculations.

-Minor point:

Please identify the study type in the methods (abstract and manuscript).

Please clarify that necessity and concerns are the BMQ parts in the method-abstract.

Please add OR values to the results section in the abstract

Line 120: The sentence is not clear.

The abbreviation of Hemoglobin A1c% is HbA1c%, please modify the manuscript accordingly.

In table 2 footnotes, please add the CCBs abbreviation clarification.

Line 26: add a coma after information.

Line 33: add “a” before significant predictor.

Line 199: Remove “a” before studies.

Line 204: remove “on”

Reviewer #2: Several minor errors in grammar/English were in the paper. There were also some long paragraphs (eg introduction is a single paragraph). Please proofread.

Refer to 'subjects' as 'participants' throughout manuscript.

For ease, please also add the response rate as a percentage.

In the Study site and subjects section, information on data collection was included. This should be placed in a separate Procedure section. Interviews are referred to (line 82), but the instruments used were questionnaires. Does this mean that the questions were read to participants? If so, this needs to be acknowledged in the limitations in the Discussion section.

Add references to the English and Arabic versions of the BMQ in the section starting on line 96.

Explain why the percentages for medications in Table 2 add up to more than 100% (presumably this means that participants were taking more than one medication for their diabetes, but this needs to be clarified).

The Discussion section is well-written and evaluates the literature well. A minor issue is the use of reference 27 which is a survey of doctors. Isn’t there any evidence-based research on the consequences of non-adherence? Opinion isn’t the same as evidence.

Reviewer #3: Dear Editor,

Thank you for giving me the chance to review this manuscript. The manuscript has some issues, which I report below. Thank you.

• Line 32: “Results: “… About half of the participants (46.5%) were moderate adherents”

I believe the study focuses on medication non-adherence, as evident in the title, so the non-adherence rate should be reported in the results in the abstract unless the author wishes to change the title for consistency.

• Line 33: “Higher necessity mean was significant predictor of high adherence”:

This should be modified to something like: necessity score was a significant predictor of adherence----similar modification is needed for concern score and frequency.

• Line 47 :”…it's expected that the number of diabetic patients will reach 592 47 million by 2035.”

The year the author referred to here is the wrong one; please correct.

• Line 64”….. These factors include sex, beliefs about 64 medications, disease duration, and education level..”

Reword this to refer to “ the patient”. e.g, patient’s gender…..

• Line 74: “NCDEG is one of the largest 74 diabetes clinics in Jordan and its’ visitors exceed 324,000 per year”

Rewording is needed

• Line 84: …”Study ethical approvals were 84 obtained from Al-Zaytoonah ethical committee and NCDEG committee.”

More details about the number of ethical approval are needed here

• Line 105

Please also cite the study that validated the Arabic version of the 4-IMAS

• Line 147 “…highest mean in Necessity statements was for “Without my medicines I would become very ill” (3.61±0.85), and the lowest mean was for “My life would be impossible without my medicines” (3.46±0.92).”

I don’t think such details add value, especially that the highest and lowest means are very close to each other same for the concerns statement

• Line 130

The line shows an abrupt start of the results section, referring to Table one. I suggest a smoother introduction of the result section, talking about the number of participants, response rate,….

• Line 177 : The authors wrote, “Among this study’s participants, 12.2% were low adherent, this percentage is lower than the ones reported among the diabetic patients’ in studies conducted in Nigeria[22], USA[23], Palestine[24], and Saudi Arabia[25]. However, there is still a room for improvement as other studies had reported better adherence levels[20,26].”

What room for improvement they meant? The sentence does not add up; please modify by comparing your non-adherence rates with other studies justifying the difference

• Line 183 “…Therefore, moderate adherence among diabetic patients also should be improved.”

Why moderate? Where did you conclude this? Did you mean both moderate and poor adherence levels? Then you better clarify this.

• Line 194: “…..However, none of these variables was significantly associated with 195 medication adherence in the present study.”

The study did not explore ethnicity or comorbidities as predictors of adherence, so the statement isn't valid

• Line 202: “…Better patients’ medication beliefs, represented as 202 increased perception of the medications’ necessity, was associated with improved medication 203 adherence.”

Were the authors here referring to their results? Please specify. This is seen in multiple places throughout the manuscript.

• Line 239: “The current study findings show a good margin for medication adherence improvement among patients with type 2 diabetes.”

I do not see 12% non adherence as a good margin; please rephrase the conclusion.

• Some English language issues and some terminology needs to be revised through the text such as “low educated” which should be replaced with: low levels of education”…..

6. PLOS authors have the option to publish the peer review history of their article (what does this mean?). If published, this will include your full peer review and any attached files.

Reviewer #1: No

Reviewer #2: No

Reviewer #3: **Yes: **Eman Alefishat

---

## [Author Response · Author response to Decision Letter 0]

18 Jul 2021

Dear Editor,

We would like to thank the editor and reviewers for their efforts and time. We appreciate the reviewers’ comments, and we believe that their comments significantly improved the quality of the manuscript. We believe that we have addressed all the comments raised by the reviewers and the editor.

Regards

Editor Comments: The paper should be checked by a professional speaker of English before complete acceptance.

- Thank you for this, the manuscript has been revised by a native English speaking professional. 

Reviewer #1: Comments to editor and authors

Thank you for submitting this interesting manuscript that that evaluated the factors associated with non-adherence among diabetic Jordanians. The study is well written however, I have few comments that should be addressed.

-Major point:

Add sample size calculations.

- Thank you for your comments. Sample size calculation has been added as suggested.

-Minor point:

Please identify the study type in the methods (abstract and manuscript).

- The study type has been added.

Please clarify that necessity and concerns are the BMQ parts in the method-abstract.

- Thank you for your comment. The abstract has been modified accordingly. 

"The four-item medication adherence scale (4-IMAS) and the beliefs about medications ‎questionnaire (BMQ) which includes necessity and concerns were used."

Please add OR values to the results section in the abstract

- The ORs were added.

Line 120: The sentence is not clear.

- The sentence has been deleted.

The abbreviation of Hemoglobin A1c% is HbA1c%, please modify the manuscript accordingly.

- Thank you for this; the abbreviation was corrected throughout the manuscript.

In table 2 footnotes, please add the CCBs abbreviation clarification.

- The abbreviation was added.

Line 26: add a coma after information.

- The coma was added.

Line 33: add “a” before significant predictor.

- The “a” was added.

Line 199: Remove “a” before studies.

- The “a” was removed.

Line 204: remove “on”

- The “on” was removed.

Reviewer #2: Several minor errors in grammar/English were in the paper. There were also some long paragraphs (eg introduction is a single paragraph). Please proofread.

- Thank you for this, the manuscript has been revised by a native English speaking professional. 

Refer to 'subjects' as 'participants' throughout manuscript.

- The manuscript was modified accordingly.

For ease, please also add the response rate as a percentage.

- The response rate percentage was added.

In the Study site and subjects section, information on data collection was included. This should be placed in a separate Procedure section. Interviews are referred to (line 82), but the instruments used were questionnaires. Does this mean that the questions were read to participants? If so, this needs to be acknowledged in the limitations in the 

- A separate section of the procedure has been added and the limitation section modified. 

“Furthermore, as the questionnaires were completed by the interviewer based on the participants’ ‎responses, the study results are subjected to social desirability and interviewer bias, However, in order to reduce the effect of the interviewer bias each question was read exactly as it appears to the participants without paraphrasing or interpretation.”‎

Discussion section.

Add references to the English and Arabic versions of the BMQ in the section starting on line 96.

- Thank you for this. The references have been added.

Explain why the percentages for medications in Table 2 add up to more than 100% (presumably this means that participants were taking more than one medication for their diabetes, but this needs to be clarified).

- Thank you for this. This is now explained in the results and the following was added: 

“Many of the participants were taking more than one type of antidiabetic medication and Metformin was the most frequently taken medication (86.4%),… “

The Discussion section is well-written and evaluates the literature well. A minor issue is the use of reference 27 which is a survey of doctors. Isn’t there any evidence-based research on the consequences of non-adherence? Opinion isn’t the same as evidence.

- Thank you for this. We agree with the reviewer and the following was added to the sentence to clarify that it was only doctors’ opinions

“a French crowdsourcing study reported that according to the medical doctors’ opinions”

Reviewer #3: Dear Editor,

Thank you for giving me the chance to review this manuscript. The manuscript has some issues, which I report below. Thank you.

• Line 32: “Results: “… About half of the participants (46.5%) were moderate adherents”

I believe the study focuses on medication non-adherence, as evident in the title, so the non-adherence rate should be reported in the results in the abstract unless the author wishes to change the title for consistency.

- Thank you for this. The percentage of low adherents has been added to the abstract.

• Line 33: “Higher necessity mean was significant predictor of high adherence”:

This should be modified to something like: necessity score was a significant predictor of adherence----similar modification is needed for concern score and frequency.

- The abstract was modified accordingly. 

“The significant predictors of the adherence level were necessity score (OR=14.86, p-value <0.01), ‎concern score (OR=0.36, p-value <0.05), and frequency of medication administration (OR=0.88, ‎p-value <0.01)."

Line 47 :”…it's expected that the number of diabetic patients will reach 592 47 million by 2035.”

The year the author referred to here is the wrong one; please correct.

- Thank you for pointing this out. The reference that contained the correct year was added and the previous reference was removed.

• Line 64”….. These factors include sex, beliefs about 64 medications, disease duration, and education level..”

Reword this to refer to “ the patient”. e.g, patient’s gender…..

- The sentence was reworded. 

(These factors include patients’ sex, their beliefs about medications, disease duration, and ‎education level.)

• Line 74: “NCDEG is one of the largest 74 diabetes clinics in Jordan and its’ visitors exceed 324,000 per year”

Rewording is needed

- The sentence was reworded.

NCDEG is one of the largest diabetes clinics in Jordan providing services to more than 324,000 patients per year.)

• Line 84: …”Study ethical approvals were 84 obtained from Al-Zaytoonah ethical committee and NCDEG committee.”

More details about the number of ethical approval are needed here

- The ethical approval numbers has been added. 

• Line 105

Please also cite the study that validated the Arabic version of the 4-IMAS

- Thank you for this; a reference has now been added

• Line 147 “…highest mean in Necessity statements was for “Without my medicines I would become very ill” (3.61±0.85), and the lowest mean was for “My life would be impossible without my medicines” (3.46±0.92).”

I don’t think such details add value, especially that the highest and lowest means are very close to each other same for the concerns statement

- This sentence has been removed.

• Line 130

The line shows an abrupt start of the results section, referring to Table one. I suggest a smoother introduction of the result section, talking about the number of participants, response rate,….

- The introduction to the Results section has been modified. 

Two hundred and eighty seven diabetic patients participated in the study with a response rate of ‎‎79.7%. As shown in Table 1, the mean age of the participants was 56(±14)). ‎

• Line 177 : The authors wrote, “Among this study’s participants, 12.2% were low adherent, this percentage is lower than the ones reported among the diabetic patients’ in studies conducted in Nigeria[22], USA[23], Palestine[24], and Saudi Arabia[25]. However, there is still a room for improvement as other studies had reported better adherence levels[20,26].”

What room for improvement they meant? The sentence does not add up; please modify by comparing your non-adherence rates with other studies justifying the difference

- Thank you for this. Reference to a lower rate of non-adherence reported in another study has been added and justification for different results was included.

However, there is still a room for improvement as, for instance, a study conducted in Ghana ‎reported a percentage of 8% of low adherence level among its’ participants‎. These variation in reported adherence levels between different studies may be attributed to differences in methodologies (i.e., questionnaires used) and sample characteristics

• Line 183 “…Therefore, moderate adherence among diabetic patients also should be improved.”

Why moderate? Where did you conclude this? Did you mean both moderate and poor adherence levels? Then you better clarify this.

- The sentence was clarified. 

Therefore, in addition to poor adherence, moderate adherence among diabetic patients also ‎should be improved.

• Line 194: “…..However, none of these variables was significantly associated with 195 medication adherence in the present study.”

The study did not explore ethnicity or comorbidities as predictors of adherence, so the statement isn't valid

- Reference to ethnicity and comorbidities has been removed. 

• Line 202: “…Better patients’ medication beliefs, represented as 202 increased perception of the medications’ necessity, was associated with improved medication 203 adherence.”

Were the authors here referring to their results? Please specify. This is seen in multiple places throughout the manuscript.

- The sentence was modified.

In the current study, better patients’ medication beliefs, represented as increased.

• Line 239: “The current study findings show a good margin for medication adherence improvement among patients with type 2 diabetes.”

I do not see 12% non adherence as a good margin; please rephrase the conclusion.

- The conclusion has been rephrased.

The current study findings show that there is area for medication adherence improvement ‎among patients with type 2 diabetes.

• Some English language issues and some terminology needs to be revised through the text such as “low educated” which should be replaced with: low levels of education”…..

- The sentence has been modified. 

which could be developed due to medication non-adherence should be explored for patients with ‎low level of education in particular during the delivery of disease management intervention ‎programs

- The whole manuscript has been revised by a native English speaker who is an academic.

Changes on references:

- Reference two (Saeedi P, Petersohn I, Salpea P, Malanda B, Karuranga S, Unwin N, et al. Global and ‎regional diabetes prevalence estimates for 2019 and projections for 2030 and 2045: ‎Results from the International Diabetes Federation Diabetes Atlas, 9th edition. Diabetes ‎Res Clin Pract [Internet]. 2019 Nov 1 [cited 2021 Jan 7];157. Available from: ‎https://pubmed.ncbi.nlm.nih.gov/31518657/‎) was replaced with (Guariguata L, Whiting DR, Hambleton I, Beagley J, Linnenkamp U, Shaw JE. Global ‎estimates of diabetes prevalence for 2013 and projections for 2035. Diabetes Res Clin ‎Pract. 2014 Feb;103(2):137–49. ‎)

- References for BMQ (References 17 and 18) were added and the reference for the Arabic version of 4-IMAS was added (Reference 20).

---

## [Editor Report · Decision Letter 1]

12 Aug 2021

Exploring variables associated with medication non-adherence in patients with type 2 diabetes mellitus

PONE-D-21-17503R1

Dear Dr. Al-Qerem,

We’re pleased to inform you that your manuscript has been judged scientifically suitable for publication and will be formally accepted for publication once it meets all outstanding technical requirements.

Kind regards,

Muhammad Sajid Hamid Akash

Academic Editor

PLOS ONE
---

## [Editor Report · Acceptance letter]

13 Aug 2021

PONE-D-21-17503R1 

Exploring variables associated with medication non-adherence in patients with type 2 diabetes mellitus 

Dear Dr. Al-Qerem:

I'm pleased to inform you that your manuscript has been deemed suitable for publication in PLOS ONE. Congratulations! Your manuscript is now with our production department. 

Kind regards, 

on behalf of

Dr. Muhammad Sajid Hamid Akash 

Academic Editor

PLOS ONE